# Non-Convulsive Status Epilepticus in Aneurysmal Subarachnoid Hemorrhage: A Prognostic Parameter

**DOI:** 10.3390/brainsci13020184

**Published:** 2023-01-22

**Authors:** Martin Vychopen, Tim Lampmann, Harun Asoglu, Agi Güresir, Hartmut Vatter, Johannes Wach, Erdem Güresir

**Affiliations:** 1Department of Neurosurgery, University Hospital Leipzig, 04103 Leipzig, Germany; 2Department of Neurosurgery, University Hospital Bonnl, 53127 Bonn, Germany

**Keywords:** non-convulsive status epilepticus, subarachnoid hemorrhage, EEG

## Abstract

A non-convulsive status epilepticus (ncSE) is a potentially fatal complication for patients in neurointensive care. In patients with aneurysmal subarachnoid hemorrhage (SAH), ncSE remains scarcely investigated. In this study, we aim to investigate the frequency and influence of non-convulsive status epilepticus on outcome in patients with SAH. We retrospectively analyzed data of consecutive patients with aneurysmal subarachnoid hemorrhage and evaluated clinical, radiological, demographical and electroencephalogram (EEG) data. Outcome was assessed according to the modified Rankin Scale (mRS) at 6 months and stratified into favorable (mRS 0-2) vs. unfavorable (mRS 3-6). We identified 171 patients with SAH, who received EEG between 01/2012 and 12/2020. ncSE was diagnosed in 19 patients (3.7%), only one of whom achieved favorable outcome. The multivariate regression analysis revealed four independent predictors of unfavorable outcome: presence of ncSE (*p* = 0.003; OR 24.1; 95 CI% 2.9–195.3), poor-grade SAH (*p* < 0.001; OR 14.0; 95 CI% 8.5–23.1), age (*p* < 0.001; OR 2.8; 95 CI% 1.6–4.6) and the presence of DIND (*p* < 0.003; OR 1.9; 95 CI% 1.2–3.1) as independent predictors for unfavorable outcome. According to our study, development of ncSE in patients suffering SAH might correlate with poor prognosis. Even when medical treatment is successful and no EEG abnormalities are detected, the long-term outcome remains poor.

## 1. Introduction

Subarachnoid hemorrhage (SAH) is a devastating disease with high mortality and morbidity despite improvement of treatment strategies and management [1]. To improve outcome, it is necessary to identify potential factors that additionally aggravate the clinical condition of the patients beside the severity of SAH. Apart from delayed cerebral ischemia (DCI) and hydrocephalus, epileptic seizures represent a severe complication in the course of the SAH therapy [2]. Reports on late epileptic seizures in SAH, defined as those occurring between 24 h and 6 weeks after SAH onset, showed an association with early re-bleeding before aneurysm treatment [3]. In the same study, the onset of seizures was identified as an independent risk factor for unfavorable neurological outcome.

In case of an initial treatment failure of the epileptic seizure, the patient might develop non-convulsive status epilepticus (ncSE). ncSE is reported to be a severe and potentially outcome-limiting complication in neurointensive-care patients [4]. Reports on ncSE in SAH patients are sparse. Recently, Kikuta et al. [5] reported on 66 patients, who underwent clipping due to aneurysmal SAH. In this series, only 10 patients developed ncSE in the course of the aSAH therapy. To date, there is no general consensus regarding the effect of the seizure according to the severity of the SAH [5,6].

Isolated epileptic seizures have been previously correlated with secondary brain damage. Therefore, the detection and corresponding treatment of ncSE might be of upmost importance to improve the outcome [1,7,8], particularly in unconscious, non-sedated patients [2,4].

The aim of the present study was to investigate the frequency for developing ncSE, as well as the influence of ncSE on outcome in a retrospective SAH cohort.

## 2. Materials and Methods

### 2.1. Patients

Between January 2012 and December 2020, 506 patients suffering from spontaneous aneurysmal SAH were admitted to our institution. SAH was diagnosed by computer tomography (CT) or lumbar puncture. All patients with spontaneous SAH underwent four-vessel digital subtraction angiography (DSA) to identify the source of bleeding. We excluded all patients with traumatic origin of SAH. Clinical data, including patient characteristics on admission and during the treatment course, radiological features (including Fisher scale) and functional neurological outcome were collected and entered into computerized database (IBM SPSS Statistics for Windows, Version 27.0., Armonk, NY, USA: IBM Corp.). The World Federation of Neurological Surgeons (WFNS) scale was used to grade patients on admission [9]. WFNS grades 1–3 were considered “good grade SAH”, and WFNS grades 4–5 were considered “poor-grade SAH”.

### 2.2. Clinical Management

We followed an early treatment strategy (within 24–48 h) in patients suffering from SAH in all clinical grades [10,11,12]. All patients who underwent aneurysm treatment received CT scan to rule out procedure-related complications. External ventricular drains were inserted in case of an acute hydrocephalus based on admission CT scan. Exceptions were made in case of comatose patients who routinely received external ventricular drain to provide an adequate ICP monitoring because of persistent impairment of consciousness. [13]. All patients received nimodipine from the day of admission to hospital. Screening for cerebral vasospasm (CVS) was performed daily using neurological examination and transcranial Doppler ultrasound (TCD) measurements. If CVS was suspected based on TCD or on delayed ischemic neurological deficit (DIND), CT angiography/perfusion (CT-A, CT-P) was performed to confirm CVS. In cases of onset of clinically relevant CVS, hypertension was induced with catecholamines during the treatment course [14]. Furthermore, on day 6–8 after SAH, all patients received a routine CT angiography/perfusion (CT-A, CT-P) to rule out possible delayed cerebral ischemia (DCI). In case of neurological deterioration, a CT-A/CT-P was performed any time necessary [15].

Delayed cerebral ischemia (DCI) was defined as occurrence of new infarction on any radiological imaging during treatment course compared to the initial CT scan on admission [16].

### 2.3. Outcome Assessment

The primary outcome measure of the present study was the neurological functional status 6 months after initial bleeding. Outcome was assessed according to the modified Rankin Scale (mRS) and stratified into favorable (mRS 0–2) and unfavorable (mRS 3–6).

We subsequently divided patients into following groups: no seizure, isolated seizure, ncSE and CSE. Then, we analyzed the outcome and mortality on discharge day.

### 2.4. Detection of Epileptic Seizure

All patients with good grade SAH, who either failed to reach extubation due to persisting neurological impairment or clinical deterioration, which could not be adequately explained on CT-A or CT-P imaging, underwent serial electroencephalography (EEG).

The diagnosis of isolated seizure was based on clinical symptoms by the treating physicians. The diagnosis of ncSE was based on consciousness impairment and positive findings on EEG and cEEG [17,18]. The diagnosis of convulsive status epilepticus (CSE) was defined as prolonged generalized convulsive seizures (>5 min) [19].

When epileptic seizures were detected, anti-epileptic monotherapy with levetiracetam was initiated. In case of persistent poor neurological status, continuous EEG (cEEG) was used for extended monitoring. The results of the cEEGs were evaluated on daily basis by a multidisciplinary board including certified neurologist, anesthesiologist and neurosurgeon.

In patients with persisting ncSE despite the initial monotherapy, we subsequently added lacosamide and brivaracetam to the anti-epileptic drug (AED) therapy. In case of further persistence of ncSE despite therapy escalation, we induced deep sedation state with isoflurane and aimed to achieve burst suppression according to our intensive-care algorithm [4,20]. Subsequently, we widened the AED therapy according to the multidisciplinary consensus for each patient individually. In the present cohort, up to 5 AEDs were used in combination to treat super-refractory ncSE.

### 2.5. Statistical Analysis

The data analysis was performed using the computer software package SPSS (version 25, IBM Corp., Armonk, NY, USA). Unpaired T-test was used for parametric statistics. Variables with non-normal distribution were analyzed with Mann–Whitney U test. Categorical variables were analyzed in contingency tables using Fisher’s exact test. Results with *p* < 0.05 in the univariate analysis were considered potentially independent predictors for poor outcome and were used to perform the multivariate analysis. A backward stepwise method was used to construct a multivariate logistic regression model with an inclusion criterion of a *p*-value < 0.05.

## 3. Results

Overall, 506 patients with aneurysmal SAH were identified between 2012 and 2020. In 171 (34%) patients, an EEG was performed due to suspected seizure. One hundred twenty-five patients (24.7%) had at least one epileptic seizure, either in the prehospital phase or during treatment at the neurointensive care unit. Of these patients, 19 (15.2%) developed ncSE and 3 (2.4%) convulsive status epilepticus (cSE). For detailed information, see Figure 1.

All patients with ncSE presented with focal epileptic activity. ncSE occurred on day 18 ± 15 days. There was no difference on the hydrocephalus rates between patients without ncSE and those with ncSE (66% vs. 74%; *p* = 0.80). ncSE treatment was successful in 14 patients (74%). All five patients with unsuccessful ncSE treatment deceased within early treatment course. Overall, only one patient with ncSE achieved favorable outcome, while nine patients achieved unfavorable outcome and nine patients deceased within 6 months after initial SAH. None of the patients who developed ncSE in the course of the treatment had a history of epileptic seizures. Five of the nineteen patients who developed ncSE had an epileptic seizure in the pre-hospital phase as the presenting symptom of the bleeding.

Table 1 presents the demographic data of the patients divided into patients with and without ncSE. We found no significant differences in patient characteristics.

Patients with ncSE were administered up to five AEDs until ncSE resolved on EEG. At discharge from NICU or at the time of therapy limitation due to unsuccessful ncSE treatment, two patients were being treated with one AED, eight patients with two AEDs, seven patients with three AEDs and two patients with five AEDs.

### 3.1. Multivariate Analysis for Outcome

The following variables showed significant correlation to outcome in univariate analysis: poor-grade SAH (*p* < 0.001), age (*p* < 0.001), ncSE (*p* < 0.001), aneurysm location (*p* < 0.001) and the presence of DIND (*p* < 0.001). All variables were used for our multivariate analysis, which showed that ncSE (*p* = 0.003; OR 24.1; 95 CI% 2.9–195.3), poor-grade SAH (*p* < 0.001; OR 14.0; 95 CI% 8.5–23.1), age (*p* < 0.001; OR 2.8; 95 CI% 1.6–4.6) and the presence of DIND (*p* < 0.003; OR 1.9; 95 CI% 1.2–3.1) were identified as independent predictors of poor outcome (Table 2).

### 3.2. Outcome According to the Presence of the Seizure

We subsequently divided the patients according to the presence of the seizure into the following groups: no seizure, isolated seizure, ncSE and CSE. The presence of isolated seizure and ncSE was accompanied by higher rates of patients with poor outcome (*p* < 0.0001). With Cramér’s V = 0.24, a middle strong correlation effect was observed. (see Table 3)

### 3.3. Fisher and Hunt and Hess Scales

We found no correlation between Fisher scale and the presence of seizure/ncSE. However, poor-grade SAH showed significantly higher probability of seizure/ncSE development. For detailed information, see Table 4.

### 3.4. Clipping vs. Coiling

We found no significant correlation between the aneurysm localization (*p* = 0.18), type of the treatment (*p* = 0.16) and development of the epileptic seizure (Table 5). The exact localization of the aneurysm according to the main vessel is presented in Appendix A.

### 3.5. Other Complications

We found nine patients with re-bleeding events after initial aneurysm treatment. Three patients underwent secondary decompressive hemicraniectomy (>48 h after initial bleeding) due to therapy-refractory elevation of intracranial pressure.

The ncSE occurred at 18 ± 15 days (IQR 7–29 days). AED combination therapy was tailored individually for each patient, and the number of AEDs was up to five AEDs per patient. In five cases, a deep sedation with isoflurane was performed as maximal escalation of the administered medication. Five patients (out of which two received burst suppression) showed persisting ncSE despite the therapy. The median duration of ncSE therapy was 9 ± 5.9 days.

Among the patients with ncSE, nine aSAH occurred on the right side, whereas seven occurred on the left side and there without a lateralization. We found no significant association between aSAH site and presence of epileptic seizures.

## 4. Discussion

The present study reveals the high impact of ncSE on outcome in patients with SAH. Overall, 3.7% of all SAH patients developed ncSE, but only one patient with ncSE achieved favorable outcome. Our study presents the largest analysis of patients suffering from aneurysmal SAH regarding ncSE with overall 506 consecutive patients included in analysis and 19 verified cases of ncSE. Our findings based on the frequency of ncSE are in accordance with the published data.

ncSE is reported to be a potentially fatal complication with high 30-day mortality and poor long-term outcome [4]. Towne et al. reported a frequency of ncSE in 8% of all comatose patients treated at NICU, who suffered traumatic brain injury, stroke, anoxia or infection, and concluded that EEG should be included in routine evaluation of comatose patients [21]. In our retrospective cohort, the ncSE diagnostic algorithm was in accordance with Towne’s recommendation. In case of ncSE suspicion, low threshold EEG and subsequent cEEG were routinely performed, and all of them underwent multidisciplinary evaluation. Therefore, our stepwise diagnostic workflow should provide high sensitivity for ncSE diagnosis.

Up to date, very few studies have examined ncSE in SAH patients. The reported frequency of ncSE in SAH varies between 3% and 14% [22,23,24,25]. In a study investigating sedated patients with cEEG, only one patient with ncSE with a duration of 5 h was identified. Solitary non-convulsive seizures were more frequent [26]. These findings accentuate the risk of epileptic activity even in sedated patients. The frequency of ncSE of patients treated on neurological/neurosurgical ICUs varies greatly in the literature. Furthermore, the cohorts for the analyses are often inhomogeneous due to the investigational nature of these studies.

Most of the studies investigate ncSE using cEEG in comatose patients solely. In the present study, all patients with either poor-grade or good-grade SAH who could not be adequately clinically monitored underwent EEG to rule out possible non-convulsive seizure. When epileptic abnormalities were detected, cEEG was applied. This should sufficiently explain the high number of patients who underwent at least one single-screening EEG for epileptic activity (*n* = 171) and the overall rather lower frequency of ncSE compared to the literature.

In accordance with the findings reported by Dennis et al., ncSE occurred on day 18 ± 15 days [23]. In five patients, ncSE treatment was unsuccessful and led to limitation of therapeutic effort at the request of the family. Overall, only one patient achieved favorable neurological outcome. This patient was also the only one presenting with a WFNS grade 1 on admission. This also corresponds to the cohort of De Marchis et al. [25], who reported that seizure burden is strongly associated with functional outcome and cognitive impairment. In our study, treatment of ncSE was unsuccessful based on cEEG monitoring in many cases, despite multidrug AED administration and even sedation with the aim to achieve burst-suppression EEG. Generally, this would support an ncSE being an additional risk factor for poor prognosis of the patient. In case of poor WFNS grade accompanied with ncSE, our data would support detailed consultation with the relatives and possibly also a restrictive therapy approach. However, this statement is strongly limited with the low number of patients and retrospective design of this study and is to be met solely individually.

Such as in TBI, there is no consensus on seizure prophylaxis, and solely prophylactic administration of AEDs in patients with aSAH remains controversial [27]. Many authors suggested that poor neurological recovery might be caused by AED-related side effects rather than by seizure activity, which would support the restrictive approach to prophylactic use of AEDs [28,29,30]. However, in a systematic review of the literature, no evidence to support or refute the use of AEDs in SAH was found [31]. Due to the scarcity of the condition, a multi-centric evaluation is necessary to answer this particular question, and the AED usage regimen remains strongly individualized.

In our study, the AED treatment was solely administered after verification and under continuous monitoring of the patient. In case of deep sedation with burst-suppression EEG, up to five AEDs were administered. However, the exact dosage and drug combination was tailored for each patient individually according to multidisciplinary consensus. After proof of absent epileptic activity on EEG and successful treatment of ncSE, AEDs were accordingly reduced to minimize the AED-related side effects. 

On the other hand, the prophylactic use of AEDs in order to prevent the development of ncSE, prolonged ICU stay and subsequent longer neurological recovery might also be discussed in individual cases. In our study, we found one patient who achieved favorable outcome, although the neurological recovery after ncSE lasted up to 6 days and meant prolonged ICU stay. Further examination of the AED algorithm for aSAH patients with ncSE should also undergo a multi-centric evaluation and goes beyond the scope of this study. 

Our results showed both isolated seizure and ncSE have a statistically significant impact on the outcome. Cramér’s V of 0.24 indicates a middle strong correlation [32] of epilepsy and outcome, which supports the hypothesis that the severity of the underlying disease, age and the presence of DCI are also independent predictors of poor outcome.

In their institutional series, Bögli et al. [6] surprisingly reported on worse outcomes in patient with lower Hunt und Hess grade (1–2) and seizure/ncSE than in cases of higher Hunt und Hess grade. This would suggest the effect of the seizure being overshadowed by the underlying severity of the SAH. Contrary to those results, Kikuta et al. report on the strong seizure effect on poor-grade SAH [5]. In our cohort, we saw a statistically significant correlation between the poor-grade SAH and the probability of seizure/ncSE. As stated by Freeman et al. [33], this would support our thesis of low-threshold continuous EEG.

ncSE is frequent following acute brain injury, and a prospective observational study of 479 patients revealed that in-hospital ncSE is independently associated with a pro-inflammatory state following aSAH. Those findings suggest that an anti-inflammatory therapy might also influence the burden of ncSE [34]. Further insights into the potential link between inflammation and seizure will be provided by an ongoing multicentric prospective randomized controlled trial investigating dexamethasone in aSAH patients [35].

### Limitations

The present study has several limitations. The data collection and statistical analysis were retrospective. The findings represent only a single-center experience. cEEG was not regularly used from admission date. cEEG was performed in case of an unexplained neurological deterioration or if poor neurological condition was persistent after epileptic activity was detected on standard EEG examination. 

As far as poor-grade SAH patients are concerned, every patient was at least once screened with EEG, and subsequently all patients with epileptic activity underwent cEEG until the successful ncSE treatment or treatment limitation. This minimizes the possibility of undetected ncSE. However, without establishing a cEEG screening from the beginning of the SAH treatment, exact assessment of the beginning of the ncSE is not possible. Despite the low-threshold diagnostics, the strong limitation of our study is the missing and unified diagnostic protocol.

Despite the low-threshold diagnostics, the limitation of our study is the fact that despite maximal effort and a tailored AED scheme with multiple substances being used, the concrete therapy and dosage were tailored for each patient individually. 

Due to our institutional algorithm, we cannot provide any risk/benefit evaluation of the prophylactic use of AEDs in SAH patients. 

## 5. Conclusions

Our findings emphasize the significant negative effect of ncSE on neurological outcome in patients suffering from SAH. Despite successful treatment of ncSE, only one patient achieved favorable outcome. This was the only patient with SAH WFNS grade 1. Those findings suggest that ncSE is associated with low rates of favorable outcome, even if the anti-epileptic treatment is considered successful. Diagnosis of ncSE in patients suffering from SAH, especially in poor-grade SAH, should be considered for consultation with the relatives of the patients to enable a maximum tailored decision-making process regarding the treatment course. The frequency and the difficulty to detect and treat ncSE demonstrate the necessity for treatment of SAH patients in high-volume neurovascular centers.

## Figures and Tables

**Figure 1 brainsci-13-00184-f001:**
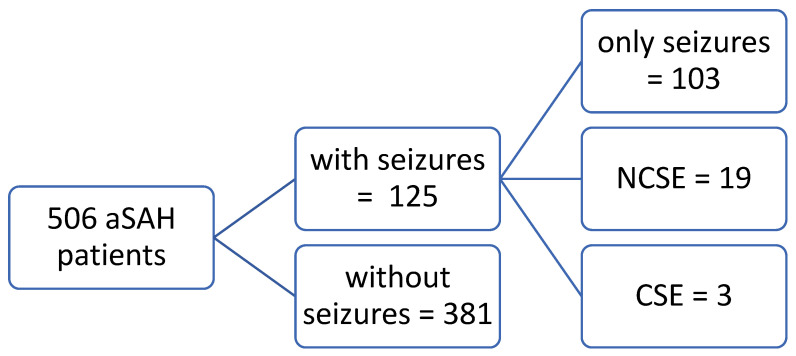
Characteristic of the cohort considering the seizure development.

**Table 1 brainsci-13-00184-t001:** Patient characteristics.

	No ncSE (*n* = 487)	ncSE (*n* = 19)	*p* Values
Age (years)	57 ± 13.7	62 ± 10.6	0.49
Female (%)	312 (64)	14 (74)	0.47
WFNS score	3 (IQR 1–5)	3 (IQR 2–5)	0.19
GCS on admission	10 (IQR 3–14)	10 (IQR 3–14)	0.84
ICH (%)	106 (22)	2 (10)	0.39
IVH (%)	158 (32)	9 (47)	0.21
Anterior-circulation aneurysm (%)	409 (84)	17 (89)	0.71
Posterior-circulation aneurysm (%)	78 (16)	2 (11)	0.71
Clipping (%)	226 (47)	9 (47)	0.53
Coiling (%)	231 (47)	10 (53)	0.53
No treatment (%)	30 (6)	0 (0)	0.53
Hydrocephalus (%)	334 (66)	14 (74)	0.80
Clinically relevant vasospasm (%)	227 (47)	9 (47)	0.65
DCI (%)	71 (15)	5 (26)	0.18
Favorable outcome (mRS 0–2) (%)	236 (49)	1 (5)	0.001

DCI—delayed cerebral ischemia; mRS—modified Rankin Score; WFNS—World Federation of Neurosurgical Societies; ICH—intracerebral hemorrhage; ICH—intraventricular hemorrhage; ncSE—non-convulsive status epilepticus; GCS—Glasgow coma scale; IQR—interquartile range.

**Table 2 brainsci-13-00184-t002:** Multivariate analysis for outcome.

	*p* Value	Odds Ratio	Confidence Interval
ncSE	0.003	24.1	2.9–195.3
Poor-grade SAH	0.001	14.0	8.5–23.1
Age DIND	0.001 0.003	2.8 1.9	1.6–4.6 1.2–3.1

Abbreviations: ncSE—non-convulsive status epilepticus; SAH—subarachnoid hemorrhage; DIND—delayed ischemic neurological deficit.

**Table 3 brainsci-13-00184-t003:** Outcome stratified by absence of seizure, isolated seizure, ncSE and CSE.

	No Seizure	Isolated Seizures (*n* = 487)	ncSE (*n* = 19)	CSE
Favorable outcome (mRS 0–2) [Percentage deviation]	274 [+15.4%]	37 [−39.9%]	1 [−92.3%]	0 [N/A]
Poor (mRS ≥ 3) [Percentage deviation]	107 [−26.4%]	66 [+75%]	18 [+127.6%]	3 [N/A]
Mortality on discharge day [Percentage deviation]	97 [+2%]	21 [−14%]	7 [+35%]	3 [N/A]

Abbreviations: N/A—not available; ncSE—non-convulsive status epilepticus; CSE—convulsive status epilepticus; mRS—modified Rankin Scale.

**Table 4 brainsci-13-00184-t004:** Correlation between Fisher grading, Hunt and Hess classification and presence of seizure/ncSE.

	No Seizure	Isolated Seizures (*n* = 487)	ncSE (*n* = 19)	*p* Value
Fisher Grade				0.58
1	20	2	1	
2	6	1	1	
3	338	98	16	
4	16	3	1	
Hunt and Hess				0.006
1–2	185	37	4	
3–5	195	67	15	

Abbreviations: ncSE—non-convulsive status epilepticus; mRS—modified Rankin Scale; CAVE: three patients with convulsive status epilepticus are not included in the table. All patients with CSE had Fisher 3 bleedings and were classified as Hunt and Hess 5, 4 and 4, subsequently.

**Table 5 brainsci-13-00184-t005:** Clipping vs. coiling.

	Anterior Circulation	Posterior Circulation	Isolated Seizure	ncSE
Clipping (*n* = 235)	221	14	58	9
Coiling (*n* = 241)	186	55	42	10

Abbreviations: ncSE—non-convulsive status epilepticus; CSE—convulsive status epilepticus; mRS—modified Rankin Scale.

## Data Availability

The data are available on request from the author (M.V.) upon reasonable request.

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
