# Peer review of "Non-Convulsive Status Epilepticus in Aneurysmal Subarachnoid Hemorrhage: A Prognostic Parameter"

_brainsci, 2023, doi:10.3390/brainsci13020184_

Round 1
Reviewer 1 Report
Authors present a retrospective study on 171 patients with subarachnoid haemorrhage (SAH) who received EEG in time frame 2012-2020 to investigate frequency of non-convulsive status epilepticus (ncSE) and its influence on outcome. Out of 19 patients (3.7%) with ncSE, only one had favorable outcome.
Low number of patients and retrospective character of the study are its drawbacks. What was the total number of patients with SAH (traumatic SAH excluded) in this time period?
Additional evaluation is needed:
What is the outcome od patients who had a seizure (SAH + epilepsy)?: less, equal or more favorable compared to the patients without seizure? Less, equal or more favorable compared to patients with ncSE? What was the outcome of patients with cSE? What was the total outcome, complication and mortality rate?
Anterior/posterior circulation aneurysms needs to be explained in detail. These are all information which are needed to get the broad perspective of the patient cohort; especially patients did not undergo extensive craniotomy (interventional patients) are of interest.
Was there an association of higher Fisher / lower HH /higher WFNS Grade to patients with seizures/ncSE? Please evaluate and report.
Please report how was the ncSE treated, when did it occur (in days) following SAH, was it more often in left/right sided pathology or no difference, and how long it took for it to break it.
For Discussion I suggest to include and comment:
Freeman WD. The Double-Edged Sword of Seizures and Nonconvulsive Status Epilepticus on Aneurysmal Subarachnoid Hemorrhage Outcomes. Neurocrit Care. 2022 Jun;36(3):699-701. doi: 10.1007/s12028-022-01490-7. Epub 2022 Apr 8. PMID: 35396642.
Bögli SY, Wang S, Romaguera N, Schütz V, Rafi O, Gilone M, Keller E, Imbach LL, Brandi G. Impact of Seizures and Status Epilepticus on Outcome in Patients with Aneurysmal Subarachnoid Hemorrhage. Neurocrit Care. 2022 Jun;36(3):751-759. doi: 10.1007/s12028-022-01489-0. Epub 2022 Apr 12. PMID: 35411540; PMCID: PMC9110510.
Reviewer 2 Report
Authors conducted a retrospective study examining the incidence of non-convulsive status epilepticus in aneurysmal subarachnoid hemorrhage patients and examined its impact on the SAH outcomes as measured by modified Rankin scale. Authors show that occurrence of non-convulsive status epilepticus in SAH patients correlated with poor outcomes.
Comments:
Definition of seizures, NCSE, and CSE should be given clearly in the methods section.
Definition of DCI – was it based only on imaging. Not evaluated clinically?
How do you define cerebral vasospasm? I do not see a fishers or modified fishers grade in the table?
Statistics – reason for backward stepwise regression compared to enter method or forward stepwise regression? What were the parameters went into the regression model? Figure 2 does not give the complete details of multivariate analysis.
Details of the comorbidity in the SAH patients?
What medications the patients were on during the EEG monitoring in ICU other than the antiepileptics and before isoflurane used for burst suppression? Any Sedation? Sedation involving GABA pathways have shown detrimental effect on the SAH outcomes.
Table 1 – No seizures (487?) Figure 1 shows 381 patients did not have seizures but the table 1 shows 487 patients did not have seizures.
What do you mean by “who could not adequately clinically monitored” in the following lines– this has to be better explained - In the 162 present study, all patients with either poor-grade or good-grade SAH who could not be 163 adequately clinically monitored underwent EEG to rule out possible non-convulsive seizure.
Reviewer 3 Report
In the current manuscript the authors present a retrospective analysis of 506 patients (171 of which received an EEG) between 2012-2020. They report on the incidence of NCSE and its impact on outcome. The manuscript is well written and of interest to a broad audience. However, I have the following comments:
Major
1. The authors mention that reports on NCSE in SAH patients are sparse. However, different recent studies describe the occurrence of NCSE in patients with SAH providing similar results when predicting outcome (incl. Kikuta et al. https://doi.org/10.1016/j.clineuro.2020.106298). I would suggest to add these to the introduction.
2. Please define the diagnostic pathway of hydrocephalus (based on ICP or imaging). Was the ICP in either group comparable?.
3. Please add how NCSE was diagnosed in this cohort (in particular as the Salzburg Criteria were only introduced in 2015)
4. How many patients received withdrawal of life sustaining therapy in either group?
5. How many patients received cEEG. Is there any data on interictal epileptiform potentials?
6. Did any patients receive primary prophylactic antiepileptic medication?
7. The NCSE occurrence day is described as 18+/-15 days which seems to be a rather large SD. Please show the range of days when NCSE was diagnosed.
Minor
1. NICU is first mentioned in line 106 without explanation of this abbreviation.
2. Would suggest to present ordinal data (i.e. WFNS, GCS etc.) using median and interquartile range for better understanding.
3. Would suggest to add the Fisher Scale to the manuscript for better comparability to other publications.
Round 2
Reviewer 1 Report
Authors have sufficiently responded to reviewer remarks.
Reviewer 2 Report
Dear Authors, Thank you for making the changes.
Reviewer 3 Report
The Authors have answered all comments raised. The manuscript is of interest to a wide audience.